

# How does agonistic behaviour differ in albino and pigmented fish?

Ondřej Slavík, Pavel Horký and Marie Wackermannová

Department of Zoology and Fisheries, Faculty of Agrobiology, Food and Natural Resources, Czech University of Life Sciences in Prague, Prague, Czech Republic

## ABSTRACT

In addition to hypopigmentation of the skin and red iris colouration, albino animals also display distinct physiological and behavioural alterations. However, information on the social interactions of albino animals is rare and has mostly been limited to specially bred strains of albino rodents and animals from unique environments in caves. Differentiating between the effects of albinism and domestication on behaviour in rodents can be difficult, and social behaviour in cave fish changes according to species-specific adaptations to conditions of permanent darkness. The agonistic behaviours of albino offspring of pigmented parents have yet to be described. In this study, we observed agonistic behaviour in albino and pigmented juvenile *Silurus glanis* catfish. We found that the total number of aggressive interactions was lower in albinos than in pigmented catfish. The distance between conspecifics was also analysed, and albinos showed a tendency towards greater separation from their same-coloured conspecifics compared with pigmented catfish. These results demonstrate that albinism can be associated with lower aggressiveness and with reduced shoaling behaviour preference, as demonstrated by a tendency towards greater separation of albinos from conspecifics.

## INTRODUCTION

Albinism is generally the result of combinations of homozygous recessive mutations from pigmented parents, and in particular, albinos are often unable to synthesize tyrosine and melatonin hormones (*Carden et al., 1998*). This disability is not only associated with red irises and light skin colouring (oculocutaneous albinism, OCA; *Carden et al., 1998*) but also with physiological, behavioural and social alterations. Some of vertebrate albinisms are indeed associated with increased levels of tyrosine and catecholamine accompanying with physiological and behavioural changes that occur during adaptation to specific conditions in caves (*Bilandžija et al., 2013*).

Information on the prevalence of terrestrial albino animals in the wild is primarily based on reports, and information on social interactions between albinos is mostly limited to studies of specially bred strains of albino rats, whose behaviour is strongly influenced by domestication (*Himmler et al., 2014*). The eyes of albino rodents show reduced adaptation to light, often leading to photoreceptor degradation (*Prusky et al., 2002*; *Refinetti, 2007*; *Marc et al., 2008*), which in turn can cause loss of vision (*Buhusi, Perera & Meck, 2005*) and movement perception (*Hupfeld & Hoffmann, 2006*), eventually leading to acrophobia

Corresponding author
Ondřej Slavík, oslavik@af.czu.cz

and/or photophobia (*Abeelen & Kroes, 1967*; *Owen, Thiessen & Lindzey, 1970*). Albino rodents have a poorer sense of smell (*Keeler, 1942*) and display lower activity levels compared with pigmented conspecifics (*Fuller, 1967*; *DeFries, 1969*). In particular, their activity is low during the day and increases during the night (*Stryjek et al., 2013*). Albino rats also spend longer periods in deep sleep (rapid eye movement, REM), especially during the dark phase (*Benca, Gilliland & Obermeyer, 1998*), and during the night, they sleep more often out of the nests relative to pigmented conspecifics (*Stryjek et al., 2013*). Albino rats are slower to inhibit the fear response and explore new objects (*Pisula et al., 2012*), and they are less effective in completing spatial tasks (*Harker & Whishaw, 2002*). For example, albino rats displayed higher hoarding activity (*Rebouças & Schmidek, 1997*), and they burrowed more slowly and constructed less complex systems of tunnels compared with wild conspecifics (*Stryjek, Modlińska & Pisula, 2012*).

Furthermore, albino vertebrates can be found in water environments, and compared with pigmented conspecifics, they display physiological and behavioural differences. For example, blind tetra *Astyanax mexicanus* (De Filippi 1853) living in caves (*Jeffery, 2001*), compared with the pigmented surface-dwelling form, display physiological adaptations to permanent darkness and limited food availability, such as greater number of taste buds (*Yamamoto et al., 2009*) and highly sensitive sensors in the lateral line (*Yoshizawa et al., 2010*; *Yoshizawa, O'Quin & Jeffery, 2013*; *Yoshizawa et al., 2014*). Such physiological adaptations appear to have resulted in a decrease in the length of sleep (*Duboué, Keene & Borowsky, 2011*), loss of schooling behaviour (*Kowalko et al., 2013*), and an evolutionary shift from fighting to food source searching, leading to the loss of hierarchy dominance and aggressiveness (*Elipot et al., 2013*). On the contrary, blind albino catfish in caves displayed agonistic behaviour (*Parzefall & Trajano, 2010*) likely reflecting their relatively large body size and bottom-dwelling form associated with stronger competition for resources. For example, the catfish *Pimelodella kronei* (Ribeiro 1907) showed a similar level of aggressiveness to its pigmented and sighted ancestor *Pimelodella transitoria* (Ribeiro 1907; *Trajano, 1991*).

Albinism in catfish often occurs in surface waters as well (*Dingerkus, Seret & Guilbert, 1991*; *Britton & Davies, 2006*; *Wakida-Kusunoki & Amador-del-Angel, 2013*; *Leal et al., 2013*). *Slavík, Horký & Maciak (2015)* described the separation of albino *Silurus glanis* (Linnaeus 1758) catfish from a group of pigmented conspecifics. The irregularity of albinos in a group of pigmented conspecifics means a guiding target for predators (*Landeau & Terborgh, 1986*; *Theodorakis, 1989*), facilitating their hunting (*Ellegren et al., 1997*), and may be a reason for exclusion of albinos from a group (*Slavík, Horký & Maciak, 2015*). However, it is not yet clear whether albinism in animals is associated with alternative social behaviour, resulting in for example, ostracism. A possible alternative behavioural display is a shift in aggression altering e.g., between domesticated albino rodents and their wild ancestors and/or between troglobites and their pigmented surface-dwelling counterparts. In the present study, we observed aggressiveness in albino and pigmented catfish *Silurus glanis* from surface waters. Considering the generally lower level of aggression observed in albino animals, we assumed that agonistic behaviour would be lower in albinos than in pigmented conspecifics.

## MATERIALS & METHODS

Albinism in catfish *Silurus glanis* L. 1758 has been commonly recorded in the wild (*Dingerkus, Seret & Guilbert, 1991*), where catfish usually occur in groups (*Boulêtreau et al., 2011*). Only juvenile catfish were used to reflect the behaviour of wild fish. These juveniles were spatially separated from adults (*Slavík et al., 2007*), and showed complex social behaviours under the experimental conditions (*Slavík, Maciak & Horký, 2012*; *Slavík et al., 2016*).

### Experimental animals

The fish used in this experiment were hatchery-reared juvenile catfish. One shoal of pigmented and one shoal of albino catfish that were unfamiliar to each other were obtained from local fish suppliers (Czech Fishery Ltd., Rybářství Hluboká and Rybářství Třeboň, Czech Republic, respectively). A total of 400 approximately equally sized fish (200 from each shoal) were transported from the hatcheries to the laboratory at four months of age. The fish were transported under stable conditions in oxygenated tanks in an air-conditioned loading space. Transport lasted approximately 2 h, and there were no observable effects on the health or mortality of the fish.

The fish were maintained in two separate holding tanks (380 L each, initial density 1.9 kg m$^{-3}$; one shoal or 200 individuals per tank) for eight weeks prior to the start of the experiment. The fish were fed food pellets ad libitum (Biomar Group, Denmark, www.biomar.com) that were distributed throughout the entire tank, providing free access to food to all individuals twice a day. The fish were maintaining under a natural photoperiod, which was the same regime they had become accustomed to in the hatchery. The water was purified using biological filters with an integrated UV sterilizer (Pressure-Flo 5000; Rolf C. Hagen Inc., www.lagunaponds.com). The water temperature and dissolved oxygen were controlled automatically (HOBO data logger; Onset Computer Corporation, Bourne, MA, USA). Fish were measured (mean 103 mm; range 90–117 mm) and weighed (mean 10 g, range 6–15 g) at the end of the experiment and removed to separate tanks to prevent mixing with unused conspecifics.

All experimental fish (400 individuals) survived. After the experiment, the fish were released under the control of the Fish Management Authorities into fish ponds with extensive production management.

### Experimental design

The experiment was conducted in the laboratory between December 2013 and January 2014. A pair of randomly selected individuals of the same colour was placed into a rectangular plastic experimental arena (36 cm long, 18 cm wide, 20 cm high) at the beginning of each trial. The arena was separated by a partition into two equal parts, and the individuals were placed on opposite sides of the arena. After an acclimation period of 1 min, the partition was removed and the behaviour of the fish was recorded for 5 min using a digital camera (GoPro Hero; GoPro, Inc., San Matteo, CA, USA). The arena was flushed out and filled with clean water after every trial. In total, 40 trials (20 pairs of albinos and 20 pairs of pigmented individuals) were conducted.
## Data analysis

In the laboratory experiment, we tested two levels of aggressive interactions among juvenile catfish, designated as aggressive or mobile displays (*Lehtonen, 2014*). Aggressive displays were further subdivided into frontal and lateral displays, and mobile displays were further subdivided into chasing and biting displays (*Hsu & Wolf, 1999*; *Dijkstra et al., 2009*). The sum of aggressive interactions, referred to as 'total aggression,' was used in further analyses (*Pauers et al., 2012*). The bottom of the experimental arena was divided into six equally sized squares (9 ×12 cm) that were used in the analyses of 'mutual distance.' Mutual distance was set as a three level class variable. A mutual distance equal to 1 meant that the individuals were in the 'same zone,' meaning that both conspecifics were in the same square. A mutual distance equal to 2 meant that the individuals were in 'adjacent zones,' meaning that conspecifics were in adjacent squares. A mutual distance equal to 3 meant that the individuals were in the 'farthest zones,' meaning that there was one square between conspecifics. Variable 'size difference' was defined as the difference between the weights of two interacting conspecifics in an experimental arena (mean 1.9 g; range 0–6 g). For the purpose of analysing the probability of occurrence of different types of agonistic behaviour over time, a 'time grid' of 5 s was set. For every time grid value, the probability of occurrence of a particular agonistic behaviour was recorded as 1 (behaviour occurred) or 0 (behaviour did not occur). The EthoLog software program (http://www.ip.usp.br/docentes/ebottoni/EthoLog/ethohome.html) was used to assign the number of particular agonistic behaviours as well as the 'duration' (in seconds) that conspecifics spent at particular mutual distances.

## Statistical analysis

Statistical analyses were performed using the SAS software package (version 9.2; SAS Institute Inc., Cary, NC, USA). When necessary, the data were square root transformed to meet normality requirements.

Total aggression and duration were analysed using mixed models with random factors (PROC GLIMMIX with Poisson distribution for total aggression and PROC MIXED with normal distribution for duration). Random factors were used to account for repeated measures collected for the same experimental units (pair of conspecifics) across the duration of the experiment. The significance of each exploratory variable (i.e., fixed effects, including their interactions) in the particular model was assessed using an F-test in which we sequentially dropped the least significant effect, beginning with the full model (backward selection procedure). Least-squares means (LSM), henceforth referred to as "adjusted means," were computed for class variables. The differences between the classes were tested using a *t*-test, and a Tukey–Kramer adjustment was used for multiple comparisons. The degrees of freedom were calculated using the Kenward–Roger method (*Kenward & Roger, 1997*).

The probabilities of occurrence for particular agonistic behaviours were analysed using the generalized estimating equation (GEE) approach (*Liang & Zeger, 1986*) for categorical, repeated measurements using the GENMOD procedure with binomial distributions. This approach is an extension of generalized linear models that provides a semi-parametric approach to longitudinal data analysis. In this study, four separate GENMOD procedures

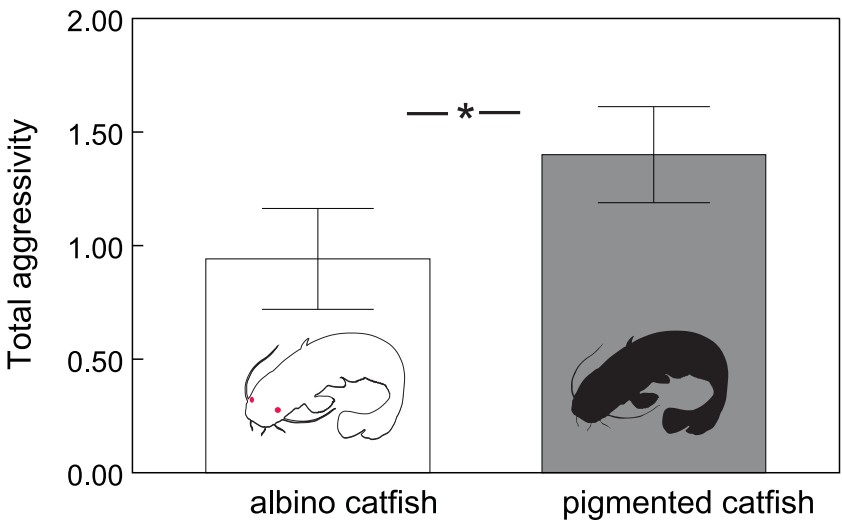

**Figure 1 Total number of aggressive interactions (adjusted means ± SE) across albino and pigmented treatments.** Significant differences are indicated (*; Tukey–Kramer adjusted $P < 0.0002$).

were designed to estimate the probability of occurrence of particular agonistic behaviours (i.e., chasing, biting, frontal and lateral displays) across the duration of the experiment.

### Ethics statement

All of the laboratory experimental procedures were in compliance with valid legislative regulations (law no. 246/1992, § 19, art. 1, letter c). The permit was granted to O. Slavík, according to Law no. 246/1992, § 17, art. 1; permit no. CZ00167. All laboratory samplings were conducted with the permission of the Departmental Expert Committee for Authorization of Experimental Projects of the Ministry of Education, Youth and Sports of the Czech Republic (permit no. MSMT-31220/2014-6). This study did not involve endangered or protected species.

## RESULTS

In total, we observed 1208 aggressive interactions, 68% of which were classified as lateral displays, 16% as frontal displays, 11% as chasing displays and 5% as biting. The total number of aggressive interactions was lower in the albino group ($F_{1, 110.6} = 14.51$, $P < 0.0002$; Fig. 1). In addition, the probability of chasing ($\chi^2 = 6.64$, d.f. = 2; $P < 0.0362$; Fig. 2A) and lateral display ($\chi^2 = 6.04$, d.f. = 2; $P < 0.0488$; Fig. 2B) changed over time and differed between groups. In the albino group, the probability of chasing decreased over time, whereas the probability of lateral display did not show any significant trend. In the pigmented group, the probability of chasing also decreased over time, whereas the probability of lateral display sharply increased. In neither group did the probability of frontal display or biting vary significantly over time. The results indicated that albinos were less aggressive compared with their pigmented conspecifics, which was primarily due to a higher probability of lateral display behaviours in the pigmented group.

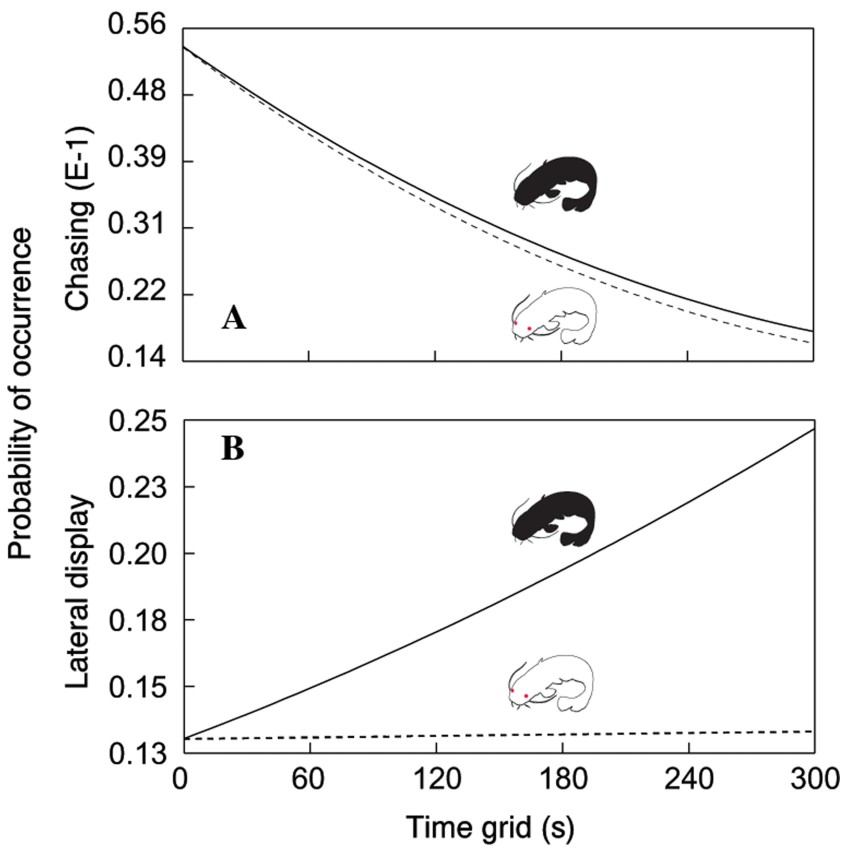

**Figure 2** **Probability of chasing (A) and lateral display (B) as a function of time across the two treatments.** Dotted line = albino catfish; black line = pigmented catfish.

Furthermore, the time that conspecifics spent at a particular distance from other conspecifics varied in both groups ($F_{5, 111} = 29.43$, $P < 0.0001$; Fig. 3). While the catfish (both albino and pigmented) generally spent the least amount of time in the farthest zones (time spent in the farthest zones did not differ between albino and pigmented catfish), albinos spent more time in adjacent zones and less time in the same zones than pigmented conspecifics (Adj. $P < 0.05$). Taken together, we found that albino catfish showed a higher tendency to be spatially separated from conspecifics, whereas pigmented catfish showed a tendency towards close contact.

## DISCUSSION

The assumption that albinism is associated with different levels of aggression is supported by this study. In particular, our results support the theory that species with different levels of colouration should display different levels of aggression (*Pryke & Griffith, 2006*) and that the level of aggression should correspond to different colour morphs (*Pryke, 2009*; *Dijkstra et al., 2009*). Indeed, one colour morph is often predicted to be more aggressive than others (*Dijkstra et al., 2010*). Our results are also consistent with recent findings showing that albinism has pleiotropic effects that are mediated through hormones that can
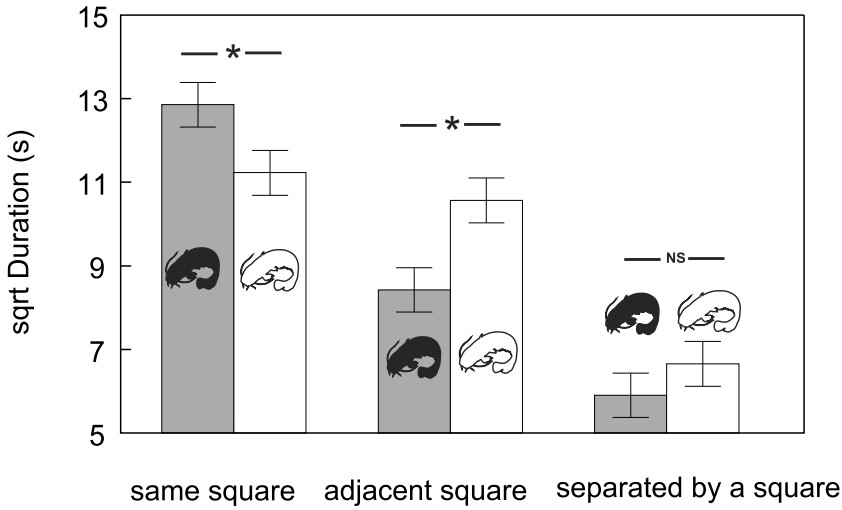

**Figure 3** **Duration (adjusted means ± SE of square root data) of time that conspecifics spent at partic-ular mutual distances across albino and pigmented treatments.** Significant differences are indicated (*; Tukey–Kramer adjusted $P < 0.05$).

affect both skin pigmentation and aggressive behaviour (*Gonzales, Varizi & Wilson, 1996*; *Ducrest, Keller & Roulin, 2008*). As reported by *Horth (2003)*, increases in the aggression of the melanic form of mosquitofish correlate with changes in melanin synthesis. Therefore, albinism, in contrast to melanism, may be generally associated with lower aggression due to shared genetic control mechanisms.

Comparison of the pigmented and albino forms of *Silurus glanis* revealed lower aggressiveness in albinos. Loss of aggressive behaviour has been reported for albinos living in caves (*Elipot et al., 2013*), and this relationship tend to be related to species-specific adaptations to unique environmental conditions. For example, most of the populations of the blind albino tetra *A. mexicanus* have lost aggressiveness, whereas individuals of the pigmented surface-dwelling form blinded in an early ontogenetic stage remain aggressive at the same level as their sighted parents (*Espinasa, Yamamoto & Jeffery, 2005*). Blind albino catfish in caves displayed similar aggressiveness to their sighted surface ancestors (*Trajano, 1991*). According to *Espinasa, Yamamoto & Jeffery (2005)*, aggressive behaviour is activated by non-optical releasers, and the reduction of aggressiveness is not the exclusive evolutionary pathway for blind albino troglobites. Loss of vision in albino cavefish *A. mexicanus* is accompanied by the development of non-visual sensors, such as neuromasts along the lateral line (*Yoshizawa et al., 2010*; *Yoshizawa, O'Quin & Jeffery, 2013*; *Yoshizawa et al., 2014*). Similarly, catfish are not typical visual predators such as salmonids (*Valdimarsson & Metcalfe, 2001*), but are adapted to prey detection in lowland rivers with a high level of turbidity, where prey are often hunted during flash floods (*Slavík et al., 2007*). Six robust tactile bristles are used for prey detection in these catfish, and with the aid of highly sensitive lateral line, the species can detect hydrodynamic traces as long as 10 s after the passage of prey (*Pohlmann, Grasso & Breithaupt, 2001*). Provided that the vision of pigmented catfish does not represent the main tool for prey detection,

it can be inferred that its role in the aggressive behaviour of albino catfish *S. glanis* is also minor. Although it can be assumed that albino catfish from surface waters are able to see, their vision may be impaired. Accordingly, albinism in mice has been correlated with acrophobia, photophobia and lower visual acuity (*Owen, Thiessen & Lindzey, 1970*; *Prusky et al., 2002*; *Buhusi, Perera & Meck, 2005*). Moreover, specially bred strains of albino rats (Sprague-Dawley) displayed a higher probability of playful attacks compared with wild-type pigmented strains (*Himmler et al., 2014*). The behaviour of this Sprague-Dawley strain, however, was also different from other albino strains. Interestingly, the authors attributed these differences to differing levels of domestication in each of the strains. Playful attacks are associated not only with domestication (see review *Himmler et al., 2014*) but also with reduced aggression, as albino rats are less aggressive than their wild-type pigmented counterparts (*Barnett, Dickson & Hocking, 1979*; *Barnett & Hocking (1981)*). Although a direct comparison between aggressiveness in catfish and rats is not possible, we speculate that albinism may be generally associated with lower aggression compared with normally pigmented conspecifics.

Aggression is also associated with social position or rank (*Mazur & Booth, 1998*; *Staffan, Magnhagen & Alanärä, 2002*). For example, an albino female vampire bat *Desmondus rotundus* bred with pigmented individuals hold the lowest social position (*Uieda, 2001*). Hence, albinism may be associated with not only lower aggressiveness but also lower dominance, as these characteristics are often correlated (*Dijkstra, Seehausen & Groothuis, 2005*; *Pryke & Griffith, 2006*). Indeed, consistent with this idea, ostracism of albino catfish by a group of pigmented conspecifics has been described (*Slavík, Horký & Maciak, 2015*), and the low ability of albinos to remain within a group may be another reason for the high predation risk in albinos (*Ellegren et al., 1997*).

Albino catfish also showed a greater tendency towards spatial separation compared with their pigmented conspecifics, which preferred to be nearer to one another. This finding differs from what was observed in domesticated albino rats, which were found to be more tolerant of conspecifics compared with wild-type strains (*Himmler et al., 2013*; *Himmler et al., 2014*). Indeed, colour-assortative shoaling is often observed (*McRobert & Bradner, 1998*; *Spence & Smith, 2006*; *Goméz-Laplaza, 2009*; *Rodgers, Kelley & Morell, 2010*). Considering the fact that albino catfish are unable to darken their body colour to avoid aggressive interactions with dominant conspecifics (*O'Connor, Metcalfe & Taylor, 1999*; *Höglund, Balm & Winberg, 2000*), spatial separation may be a strategy for avoiding the escalation of aggressive behaviours. On the other hand, if albino catfish have poor vision, then their low tendency towards grouping may be a result of this weakened physiological condition. Accordingly, blind cave fish displayed loss of schooling behaviour (*Parzefall & Trajano, 2010*; *Kowalko et al., 2013*). In the case of troglobites, however, a low tendency towards grouping is considered to represent an evolutionary adaptation to sparse prey and low food availability, conditions where life in a group is not beneficial to better foraging activity such as it is in surface waters (*Griffith et al., 2004*; *Ward & Hart, 2005*). Large numbers of albinos existing together in the wild has only been reported in insects (*Hoste et al., 2003*), and whether albinos are mutually attracted to each other and form larger groups in nature remains unknown.

## CONCLUSIONS

Similarities to the shift in the behaviour of albino catfish towards lower aggressiveness can be found in domesticated albino rodents and their wild pigmented counterparts as well as between blind cave fish and their sighted ancestors from surface waters. Therefore, loss of pigmentation may not only be linked to aggression in albinos, but also have other pleiotropic effects that can be observed, for example, as impaired eyesight in surface environments and/or specie-specific evolutionary adaptations to conditions of permanent darkness. In addition, albinos were found to maintain greater distances between themselves compared with pigmented individuals in the present study, corresponding to the loss of schooling behaviour in blind cave fish.

## ACKNOWLEDGEMENTS

The authors sincerely thank Sergio Pellis, Tobias Backström and the anonymous referee for critical evaluation and valuable comments on the manuscript. In addition, the authors wish to thank A Slavikova for the help with earlier versions of the manuscript.

### Funding

This work was supported by the Czech Science Foundation (No. 16-06498S). The funders had no role in study design, data collection and analysis, decision to publish, or preparation of the manuscript.

### Grant Disclosures

The following grant information was disclosed by the authors:
Czech Science Foundation: 16-06498S.

### Competing Interests

The authors declare there are no competing interests.

### Author Contributions

- Ondřej Slavík conceived and designed the experiments, performed the experiments, wrote the paper, prepared figures and/or tables, reviewed drafts of the paper.
- Pavel Horký conceived and designed the experiments, performed the experiments, analyzed the data, contributed reagents/materials/analysis tools, wrote the paper, prepared figures and/or tables, reviewed drafts of the paper.
- Marie Wackermannová analyzed the data.

### Animal Ethics

The following information was supplied relating to ethical approvals (i.e., approving body and any reference numbers):

All of the laboratory experimental procedures were in compliance with valid legislative regulations (law no. 246/1992, § 19, art. 1, letter c). The permit was granted to O. Slavík,

according to Law no. 246/1992, § 17, art. 1; permit no. CZ00167. All laboratory samplings were conducted with the permission of the Departmental Expert Committee for Authorization of Experimental Projects of the Ministry of Education, Youth and Sports of the Czech Republic (permit no. MSMT-31220/2014-6). This study did not involve endangered or protected species.

## Data Availability

The raw data has been supplied as Supplemental Datasets.

## Supplemental Information

Supplemental information for this article can be found online at http://dx.doi.org/10.7717/peerj.1937#supplemental-information.

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
