# Peer review of "How does agonistic behaviour differ in albino and pigmented fish?"

_PeerJ, doi:10.7717/peerj.1937_

## Round 0.1 · original submission · Major Revisions

I now have 3 reviews back on your manuscript, and all are relatively positive about the value of the study and the suitability of the manuscript for publication. However, each also has some suggestions for improvement of the manuscript that I find myself in agreement with from my own reading of the paper. In particular, we are all at a loss to explain the discrepancies in the manuscript about whether your hypothesis was that albino catfish should be more or less aggressive than their pigmented counterparts and exactly what your results support – each of the referees comment on some aspect of this because the text is either unclear or goes both ways in different places. The text needs to be revised to remove such confusion for future readers, and I agree with the suggestions about expanding the introduction and laying out a more direct logical basis for your argument that “higher levels of aggression would be observed in albino catfish compared with pigmented conspecifics” would help readers. Also, one of the reviewers noticed some ‘social commentary’ in the article (for example “albinos in human populations are generally stigmatized, often considered by local religions to be a punishment for ancestral sins or portrayed in Hollywood films as murderous villains”) - I agree with them that these types of comments seem inappropriate for an article about fish behavior, and so I suggest you delete them.

The most substantial comment from the reviewers is that the alternative hypothesis of vision should be addressed in the manuscript. This referee makes some very good points about the interaction of vision and aggression in fishes, and suggests additional experiments to differentiate between the alternatives. While I agree it would be extremely valuable to see these additional experiments to support the conclusions of the paper, I leave it to the authors how to address this comment: you could either add the additional experiment to support your conclusions as stated currently, or you could soften the conclusion of the causal role of pigmentation and include a discussion of the interaction of vision and pigmentation in the discussion of the paper (incorporating some of the list of references that were provided by the referee) to address this issue within the manuscript without additional experiments.

Because the recommendations include either revision of the conclusions and a new section of the manuscript or inclusion of additional experiments, we consider this a major revision.

·

Basic reporting

For the most part, the English expression is clear. However, there are some sentences that need to be changed or explained for the sake of greater clarity. These are as follows:

Line 30: What is a “hart”?

Line 51: Change “…they sleep more often out of the nets…” to “…they sleep more often out of the nests…”

Line 53 (and line 318): Change “Whislaw” to “Whishaw”

Line 55: Change “…burrowed slower…” to “…burrowed more slowly…”

Line 57: “…less interested…” is ambiguous as it is not clear whether it is the albino animals that are less interested or it is the potential pigmented partners that are less interested. Clarify

Line 210: Change “…breed with pigmented individuals…” to “…that bred with pigmented individuals…”

Line 220: Change “…and carrier opportunities…” to “…and career opportunities…”

I had difficulty interpreting Figure 3 as the caption did not provide information on what the different columns represent. From the description of the figure provided in the text (lines 178-183), it would be inferred that the grey bars on the left are for time spent in the same square, the white columns on the left are for time spent furthest apart and the hatched columns in the middle are for time spent in adjacent squares. This means that going from left to right, inter-animal distance is decreasing. Correct? Given that the main point of these data is showing the counter-intuitive findings that the albino catfish keep further apart than the pigmented ones, I think it would be better to display this graph differently. First, start with the smallest distance on the left and progress to the largest to the right. Second, show the distances in pairs of columns, pigmented fish on the left (black column) and albino fish on the right (white column). This would clearly show that the biggest difference between pigmented and albino fish is that the former spend more time in the same square and less time in adjacent squares, the opposite to the pattern shown by the albino fish. The X-axis should have appropriate labels (e.g., same square, adjacent squares, separated by a square).

Experimental design

The design of the experiment and the analysis of the data appear to be adequate to test the hypothesis posited in the Introduction (although see below). Moreover, the conclusions drawn from the data are consistent with the findings.

However, the theoretical framework for the hypothesis needs to be elaborated upon in the Introduction. Basically, the transition from line 57 to line 58 is not justified. In the Introduction preceding line 58, most of the discussion concerns albinism in general and an extensive description of differences in the behavior of albino rodents compared to their pigmented counterparts. Then, abruptly, on line 58, mention is made of albino fish, leading to the last sentence of the Introduction making an hypothesis about albino catfish being more aggressive than pigmented catfish. These arguments need to be elaborated upon. First, why are fish being studied? Is it simply to expand the range of animals in which albinism has been systematically studied, or, do fish, and in particular, the catfish in the present study, offer a new avenue to gain insight into albinism? Second, why is aggression the central topic of the present study? The preceding discussion dealt with many differences in the behavior of albino animals and it is not clear from that discussion as to why aggression is the most informative trait to study. This needs to be explained more fully. Third, the hypothesis posits that albino catfish should be more aggressive than pigmented conspecifics (lines 72-73), yet it is not clear to me why this should be so, especially given that in rodents, albino strains are less aggressive than pigmented strains. The basis for this hypothesis should be clarified.

Indeed, given that the findings presented in the paper show that albino catfish are less aggressive, it was surprising that the opening line of the Discussion (186-187) claims that the findings support the assumption underlying the study. The data show that albino catfish are ‘less’ aggressive than pigmented fish, not ‘more’ aggressive. This discrepancy between the hypothesis stated in the Introduction and the non-confirmatory data generated by the study should be a central focus of the Discussion. Why was the hypothesis not supported and what do the findings tell about the possible mechanisms regulating aggression? What mechanisms correlated with albinism may lead to this decline in aggressivity?

Validity of the findings

The data appear to be robust and informative

Additional comments

The design of the experiment and the analysis of the data appear to be adequate to test the hypothesis posited in the Introduction (although see below). Moreover, the conclusions drawn from the data are consistent with the findings.

However, the theoretical framework for the hypothesis needs to be elaborated upon in the Introduction. Basically, the transition from line 57 to line 58 is not justified. In the Introduction preceding line 58, most of the discussion concerns albinism in general and an extensive description of differences in the behavior of albino rodents compared to their pigmented counterparts. Then, abruptly, on line 58, mention is made of albino fish, leading to the last sentence of the Introduction making an hypothesis about albino catfish being more aggressive than pigmented catfish. These arguments need to be elaborated upon. First, why are fish being studied? Is it simply to expand the range of animals in which albinism has been systematically studied, or, do fish, and in particular, the catfish in the present study, offer a new avenue to gain insight into albinism? Second, why is aggression the central topic of the present study? The preceding discussion dealt with many differences in the behavior of albino animals and it is not clear from that discussion as to why aggression is the most informative trait to study. This needs to be explained more fully. Third, the hypothesis posits that albino catfish should be more aggressive than pigmented conspecifics (lines 72-73), yet it is not clear to me why this should be so, especially given that in rodents, albino strains are less aggressive than pigmented strains. The basis for this hypothesis should be clarified.

Indeed, given that the findings presented in the paper show that albino catfish are less aggressive, it was surprising that the opening line of the Discussion (186-187) claims that the findings support the assumption underlying the study. The data show that albino catfish are ‘less’ aggressive than pigmented fish, not ‘more’ aggressive. This discrepancy between the hypothesis stated in the Introduction and the non-confirmatory data generated by the study should be a central focus of the Discussion. Why was the hypothesis not supported and what do the findings tell about the possible mechanisms regulating aggression? What mechanisms correlated with albinism may lead to this decline in aggressivity?

Minor changes and clarifications:

Line 30: What is a “hart”?

Line 51: Change “…they sleep more often out of the nets…” to “…they sleep more often out of the nests…”

Line 53 (and line 318): Change “Whislaw” to “Whishaw”

Line 55: Change “…burrowed slower…” to “…burrowed more slowly…”

Line 57: “…less interested…” is ambiguous as it is not clear whether it is the albino animals that are less interested or it is the potential pigmented partners that are less interested. Clarify

Line 210: Change “…breed with pigmented individuals…” to “…that bred with pigmented individuals…”

Line 220: Change “…and carrier opportunities…” to “…and career opportunities…”

I had difficulty interpreting Figure 3 as the caption did not provide information on what the different columns represent. From the description of the figure provided in the text (lines 178-183), it would be inferred that the grey bars on the left are for time spent in the same square, the white columns on the left are for time spent furthest apart and the hatched columns in the middle are for time spent in adjacent squares. This means that going from left to right, inter-animal distance is decreasing. Correct? Given that the main point of these data is showing the counter-intuitive findings that the albino catfish keep further apart than the pigmented ones, I think it would be better to display this graph differently. First, start with the smallest distance on the left and progress to the largest to the right. Second, show the distances in pairs of columns, pigmented fish on the left (black column) and albino fish on the right (white column). This would clearly show that the biggest difference between pigmented and albino fish is that the former spend more time in the same square and less time in adjacent squares, the opposite to the pattern shown by the albino fish. The X-axis should have appropriate labels (e.g., same square, adjacent squares, separated by a square).

Reviewer 2 ·

Basic reporting

This manuscript entitled “How does agonistic behaviour in albino and pigmented fish differ?” written by Slavik, et al. conducted two types of behavioral observations to support the association between albinism and behavioral changes toward less aggression. There are multiple concerning points in this manuscript including one major and several minor ones. I will explain them in following sections

For the Basic Reporting,
This manuscript meets the criteria of PeerJ except following two points:
1) Authors did not provide the species names of used catfish. It must be shown in materials and methods section at least.
2) Adequate References are not presented. For example, line 77 “Bouletreau et al. 2011; Cucherousset et al., 2012” these mentioned grouping/ schooling of catfish but did not mention whether albinism is commonly seen in the group.

Experimental design

1) Not enough evidences to support their conclusion: the link between albinism and behavioral changes.
Aggressive and mobile displays and shoaling distances could highly depend on visual capacity. According to the authors, albino catfish do not have retinal pigment, which supports to form clear images on the retina (without the retinal pigment, images turn to be blur/less contrast, which affect the visual sensation of the objects). Some of vertebrate albinisms are indeed associated with physiological changes that potentially lead behavioral shifts via tyrosine-dopamine axis (as author referenced, Carden et al., 1998; and (Bilandžija et al., 2013; Grønskov et al., 2007). However, visual interaction also plays a large role in aggression and shoaling. For example,(Elipot et al., 2013; Kowalko et al., 2013; Ostlund-Nilsson et al., 2006; Plotkin, 1988). In addition, the hybrid between Mexican albino cavefish and its pigmented surface-dwelling conspecific show both less pigmentation than their parental surface-dweller yet display increased aggression comparing with surface-dweller, suggesting that pigmentation itself is not always associated with aggressiveness (Elipot et al., 2013).
To support the association, authors may run an experiment by making both pigmented and albino populations blind. For example, fish eyes can be covered by aluminum foil glued with vetbond (http://www.wpi-europe.com/products/laboratory-supplies/adhesives/vetbond.aspx or else). If the pigmentation regulates aggressiveness in this catfish, blinded pigmented and albino fish still keep distinct behaviors.

References:
Bilandžija, H., Ma, L., Parkhurst, A. and Jeffery, W. R. (2013). A potential benefit of albinism in Astyanax cavefish: downregulation of the oca2 gene increases tyrosine and catecholamine levels as an alternative to melanin synthesis. PLoS One 8, e80823.
Elipot, Y., Hinaux, H., Callebert, J. and Rétaux, S. (2013). Evolutionary shift from fighting to foraging in blind cavefish through changes in the serotonin network. Curr. Biol. 23, 1–10.
Grønskov, K., Ek, J. and Brondum-Nielsen, K. (2007). Oculocutaneous albinism. Orphanet J. Rare Dis. 2, 43.
Kowalko, J. E., Rohner, N., Rompani, S. B., Peterson, B. K., Linden, T. a, Yoshizawa, M., Kay, E. H., Weber, J., Hoekstra, H. E., Jeffery, W. R., et al. (2013). Loss of schooling behavior in cavefish through sight-dependent and sight-independent mechanisms. Curr. Biol. 23, 1874–83.
Ostlund-Nilsson, S., Mayer, I. and Huntingford, A. (2006). Biology of the Three-Spined Stickleback (CRC Marine Biology Series). Boca Raton: CRC Press.
Plotkin, H. C. (1988). The Role of Behavior in Evolution. Cambridge: The MIT Press.

Validity of the findings

The data support "albino catfish were significantly less aggressive than their pigmented conspecifics", however, there is few data to support their conclusion: albinism is MOST often associated with lower aggressiveness, lower dominance and social exclusion. Because, their working hypothesis (in introduction) was "higher levels of aggression would be observed in albino catfish compared with pigmented conspecifics" based on many other animal species, and their result was opposite, and other animal example (ex. cavefish) doesn't provide the major association between albinism and less-aggression (see above section).
They must at least test the involvement of visual ability in aggression, and reconsider their conclusion.
If authors show albinism but not visual capacity regulates less-aggression and further distance between individuals, It would be a very nice new finding adequate to be published in PeerJ.

·

Basic reporting

The significances could be included into the figures, either in the label or the actual figure.

In figure 3 definitions of the 3 different distances should be added.

Experimental design

No comments

Validity of the findings

No comments

Additional comments

The manuscript “How does agonistic behaviour in albino and pigmented fish differ?” has investigated differences between albino and normally pigmented catfish. The results indicate that albino catfish are less aggressive. The manuscript is well written and worthy of publication.

Special comments
In the abstract physiological and behavioural syndromes are mentioned but this terminology is not used anywhere else. Therefore I suggest rewording this passage in the abstract.
I also wondered if pairing of albinos and normally pigmented catfish were ever discussed? If so, any reason why it was excluded. It seems like it could be helpful for answering the hypothesis.
In the passage at line 225-234, concerning similarities with rodents. I think that the order of presenting is confusing right now. For me, it would be better with start with the behaviour of albino rodents and then state that it is similar in catfish, especially since it is not intuitive that playful attacks are negatively correlated with aggression.

Minor comments
In line 2, a comma-sign after the first author should be added.
Line 63-66. As I understand it, the references are just for the last part of the sentence. I think it is better to divide the sentence into two new sentences.
Line 80. Year is missing for Slavík et al.,.
Line 94. The word maintaining should be changed to maintained.
Line 277. Is it the novel by Dan Brown? Then Brown, D.

---

## Round 0.2 · accepted · Accept

Thank you for your revised manuscript. I have now heard back from the more critical referee who wanted to see the revised manuscript that they are satisfied with the revisions, as am I. I see no reason to delay the process any longer, and am happy to move your manuscript into production for publication at this time.

Reviewer 2 ·

Basic reporting

This Reviewed version is significantly improved and I think it meets the criteria of PeerJ.

Experimental design

It is good enough to resolve their hypothesis

Validity of the findings

their findings are new and worth to be reported